# The Teddy-Tool v1.1: temporal disaggregation of daily climate model data for climate impact analysis

Florian Zabel[1], Benjamin Poschlod[2]

[1]Ludwig-Maximilians-Universität München (LMU), Department of Geography, Luisenstr. 37, 80333 Munich, Germany
[2]Research Unit Sustainability and Climate Risks, Center for Earth System Research and Sustainability, Universität Hamburg, Grindelberg 5, 20144 Hamburg, Germany

*Correspondence to: Florian Zabel (f.zabel@lmu.de)*

**Abstract**

Climate models provide required input data for global or regional climate impact analysis in temporally aggregated form, often in daily resolution to save space on data servers. Today, many impact models work with daily data, however, sub-daily climate information is getting increasingly important for more and more models from different sectors, such as the agricultural, the water, and the energy sector. Therefore, the open source Teddy-Tool (**te**mporal **d**isaggregation of **d**ail**y** climate model data) has been developed to disaggregate (temporally downscale) daily climate data to sub-daily hourly values. Here, we describe and validate the temporal disaggregation, which is based on the choice of daily climate analogues. In this study, we apply the Teddy-Tool to disaggregate bias-corrected climate model data from the Coupled Model Intercomparison Project Phase 6 (CMIP6). We choose to disaggregate temperature, precipitation, humidity, longwave radiation, shortwave radiation, surface pressure, and wind speed. As a reference, globally available bias-corrected hourly reanalysis WFDE5 data from 1980-2019 are used to take specific local and seasonal features of the empirical diurnal profiles into account. For a given location and day within the climate model data, the Teddy-Tool screens the reference data set to find the most similar meteorological day based on rank statistics. The diurnal profile of the reference data is then applied on the climate model. The physical dependency between variables is preserved, since the diurnal profile of all variables is taken from the same, most similar meteorological day of the historical reanalysis dataset. Mass and energy are strictly preserved by the Teddy-Tool to exactly reproduce the daily values from the climate models.

For evaluation, we aggregate the hourly WFDE5 data to daily values and apply the Teddy-Tool for disaggregation. Thereby, we compare the original hourly data with the data disaggregated by Teddy. We perform a sensitivity analysis of different time window sizes used for finding the most similar meteorological day in the past. In addition, we perform a cross-validation and autocorrelation analysis for 30 globally distributed samples around the world, representing different climate zones. The validation shows that Teddy is able to reproduce historical diurnal courses with high correlations >0.9 for all variables, except for wind speed (>0.75) and precipitation (>0.5). We discuss limitations of the method regarding the reproduction of precipitation extremes, inter-day connectivity, and disaggregation of end-of-century projections with strong warming. Depending on the use case, sub-daily data provided by the Teddy-Tool could make climate impact assessments more robust and reliable.

## 1. Introduction

Sub-daily climate data is becoming increasingly important in climate impact analysis. This type of data, which captures variations in temperature, precipitation, and other weather variables at intervals of less than a day, can provide a more detailed representation of local and regional climate conditions and temporal variations. This information can be crucial for evaluating the impacts of climate change on various sectors, such as agriculture, water resources, energy production, and human health (Golub et al., 2022; Trinanes and Martinez-Urtaza, 2021; Colón-González et al., 2021; Tittensor et al., 2021; Byers et al., 2018; Jägermeyr et al., 2021; Poschlod and Ludwig, 2021; Degife et al., 2021). A better representation of the diurnal course of temperature, extreme precipitation events, and other weather variables are also important for adaptation assessments which depend on behavior or processes with high temporal dynamics, such as the energy demand, labor activity, the heat stress of crops or flood events (Minoli et al., 2022; Zabel et al., 2021; Reed et al., 2022; Orlov et al., 2021; Franke et al., 2022; Poschlod 2022). Research has shown that using sub-daily climate data can result in more robust and reliable impact assessments compared to using daily data (Orlov et al. 2023).

Today, most climate model data are available for download at daily resolution because of the high storage requirements for sub-daily climate data (Juckes et al., 2020). However, the demand for sub-daily data is increasing with future developments of data management expected to handle this demand with decreasing costs for storage and computing resources (Lüttgau & Kunkel, 2018). Different methods exist to disaggregate available daily climate data to sub-daily, most often hourly values. These can be roughly divided into statistical methods, weather generators, and mechanistic approaches, although mixed forms also exist (Förster et al., 2016).

Mechanistic methods use regional climate models to dynamically downscale atmospheric conditions in time and space, usually for a limited area (Vormoor and Skaugen, 2013; Liu et al., 2011; Kunstmann and Stadler, 2005). Weather generators generate synthetic sequences of hourly weather variables by using random number generators that match statistics (Ailliot et al., 2015; Mezghani and Hingray, 2009). Various statistical methods exist for temporal disaggregation of daily climate data, ranging from simple interpolations or deterministic approaches to non-parametric approaches and methods that derive statistical relationships from historical data or look for climate analogues (Bennett et al., 2020; Breinl and Di Baldassarre, 2019; Chen, 2016; Debele et al., 2007; Förster et al., 2016; Görner et al., 2021; Liston and Elder, 2006; Park and Chung, 2020; Verfaillie et al., 2017; Poschlod et al., 2018; Zhao et al., 2021). Each of these methods has its own advantages and limitations, and the choice of method depends on factors such as the specific needs of the impact assessment, the quality of the available data, and computational resources.

Here, we introduce the Teddy-Tool (**te**mporal **d**isaggregation of **d**ail**y** climate model data), which uses statistical methods for temporal disaggregation of daily climate model data. Existing statistical approaches are often only valid for a specific location and cannot be applied globally. In addition, available disaggregation tools often focus on only one variable (e.g. Pui et al., 2012) and therefore do not consider physical interdependencies between different variables, such as precipitation, humidity, temperature, and radiation. Teddy has been specifically developed as a globally applicable tool for climate impact studies. For this purpose, Teddy strictly preserves mass and energy of daily climate model data for each variable throughout the disaggregation procedure. Teddy additionally aims at taking regional and seasonal climate characteristics into account and considers the physical consistency between variables.

Teddy represents an easy-to-use tool that can be applied for climate impact assessments in different
sectors that allows a physically consistent temporal disaggregation of daily climate model data. The
Teddy-Tool has been written in Matlab and is available open source via Zenodo (see code availability).

## 2. Data and data requirements

In principle, the Teddy-Tool can be used with any climate input, but has specifically been developed to
be used with daily climate data for historical time periods and future scenarios from the Inter-Sectoral
Impact Model Intercomparison Project (ISIMIP). ISIMIP offers a framework for consistently projecting
the impacts of climate change across affected sectors and spatial scales (Warszawski et al., 2014). To
guarantee cross-sectoral consistency in ISIMIP, all sectors are provided with the same climate data for
historical (1850-2014) and future time periods (2015-2100) for different scenarios (SSP126, SSP370,
SSP585). ISIMIP provides bias-corrected climate model data from the Coupled Model Intercomparison
Project Phase 6 (CMIP6) and trend-preserving reanalysis climate data (Lange, 2019). Within ISIMIP,
some modeling communities from different sectors have expressed their need for sub-daily climate
data, including the agricultural and the energy sector.
Daily bias-corrected climate model data are provided by ISIMIP at 0.5° spatial resolution for air
temperature (tas), humidity (hurs), shortwave radiation (rsds), longwave radiation (rlds), air pressure
(ps), wind speed (sfcwind), and precipitation (pr) (Lange, 2019). For air temperature, the daily
maximum (tasmax) and minimum (tasmin) values are additionally provided. ISIMIP provides CMIP6
data for the climate models GFDL-ESM4, IPSL-CM6A-LR, MPI-ESM1-2-HR, MRI-ESM2-0, and UKESM1-
0-LL.
Teddy requires hourly climate data as a reference for temporal disaggregation. Therefore, we use the
WFDE5 dataset, which has been gererated using the WATCH Forcing Data (WFD) methodology applied
to ERA5 reanalysis data (Cucchi et al., 2020). The bias-adjusted hourly WFDE5 data is globally available
for the time period between 1979 and 2019 at 0.5° spatial resolution. It is consistent with the bias-
adjustment procedure within ISIMIP (Lange, 2019) and thus provides a consistent hourly reference
data for Teddy. Table 1 gives an overview of the available variables and the required datasets at their
temporal resolution. The temporal resolution of the Teddy output is adjustable by the user and can be
set to 1-, 2-, 3-, 4-, 6-, 8-, or 12-hourly values.
Table 1: Variables and units of used hourly (h) and daily (d) climate data and the Teddy output. For
WFDE5, the specific variable name is provided in brackets. WFDE5 variables have instantaneous values,
while SWdown, LWdown, Rainf and Snowf have average values over the next hour at each time step.

| Variable | WFDE5 (h) | ISIMIP Climate Model (d) | Teddy (flexible) |
|---|---|---|---|
| Air temperature (tas) | K (Tair) | K | K |
| tasmin | - | K | - |
| tasmax | - | K | - |
| Humidity (hurs/huss) | kg/kg (Qair) | % | % |
| Shortwave radiation (rsds) | $W\ m^{-2}$ (SWdown) | $W\ m^{-2}$ | $W\ m^{-2}$ |

| | | | |
|---|---|---|---|
| Longwave radiation (rlds) | W m$^{-2}$ (LWdown) | W m$^{-2}$ | W m$^{-2}$ |
| Precipitation (pr) | kg m$^{-2}$ s$^{-1}$ (Rainf+Snowf) | kg m$^{-2}$ s$^{-1}$ | mm timestep$^{-1}$ |
| Air pressure (ps) | Pa (PSurf) | Pa | hPa |
| Wind speed (sfcwind) | m s$^{-1}$ (Wind) | m s$^{-1}$ | m s$^{-1}$ |


## 3. Methods

Teddy uses an empirical approach, which 1) selects the 'most similar meteorological day' for the daily
climate model data (here: ISIMIP CMIP6 data) within the reference climate data (here: WFDE5) at the
same location. 2) Teddy applies the location-specific diurnal course to each variable of the daily climate
model data for a day of interest. In the following, the procedure is explained in detail, where the
example case of ISIMIP climate data and WFDE5 reference data is used for further illustration:
In a first precalculation step, in order to minimize computational resources, hourly WFDE5 data are
aggregated to daily values and stored as NetCDF files. The daily aggregation uses mean values for all
variables and daily sums for precipitation. In addition, rainfall and snowfall fluxes must be summed up
for WFDE5. Daily maximum and minimum temperature are calculated from the hourly data. Units of
climate inputs are converted to match the Teddy output (see Tab. 1). For the conversion of specific
humidity to relative humidity, the Buck equation is applied (Buck, 1981).After reading the daily climate
model data for the selected location (latitude/longitude) that determines a specific grid cell at 0.5°
resolution, the daily mean values of all ISIMIP variables (see Tab. 1) are compared to the aggregated
daily values of WFDE5 for a specific time step in order to identify the most similar meteorological day.
For the comparison, a day-of-year (DOY) window can be selected by the user that allows for a selection
of days around the DOY of the actual time step. By default, the DOY window size is set to 11, which
means a sequence of ± 11 days around the actual DOY. As a result, 23 days are selected from each of
the 40 WFDE5 reference years (1980-2019). These 920 days now serve as the statistical population for
further calculations (Fig. 1). In a next step, the climate model day of interest and the statistical
population of 920 WFDE5 days are classified according to their precipitation state (wet / dry). As
climate models tend to produce too many days with low-intensity precipitation called 'drizzle bias'
(Chen et al., 2021), days with aggregated daily precipitation values below 1 mm per day are considered
as dry days (Sun et al., 2006). Depending on the precipitation state of the previous day, the day of
interest and the following day, there are eight classes: dry-dry-dry, dry-dry-wet, wet-dry-dry, wet-dry-
wet, dry-wet-dry, dry-wet-wet, wet-wet-dry, and wet-wet-wet. This step is included to better
reproduce the inter-day connectivity of precipitation (Li et al., 2018). Only days with the same
precipitation class as the climate model day of interest are selected for the further course. Next, the
absolute error (AE) between daily climate model and aggregated daily WFDE5 data for each variable is
calculated for the remaining statistical population and ranked in ascending order. The ranking
approach is chosen, since the absolute or relative errors of different meteorological variables cannot
be compared to each other. The ranks are cumulated with equal weight over all variables for each day
of the statistical population. In this context, we define 'the most similar meteorological day' as the day
with the minimum sum of ranks (Fig. 1). Thus, the 'most similar meteorological day' refers to the
statistically derived similarity of all available daily near-surface meteorological variables at a given
location and time. The approach works under the assumption that similar daily values would have a
similar sub-daily profile (Li et al., 2018; Pui et al., 2012; Sharma et al., 2006). Finally, the hourly values
are taken from the most similar meteorological day of the WFDE5 reference dataset for each variable
and are divided by the WFDE5 daily mean (sum for precipitation) value of the selected day, in order to
refer to relative diurnal profiles without absolute variations (Fig. 1). The hourly profile is then applied
for each variable to the daily mean (sum for precipitation) value from the climate model. Thus, the
daily mean value (sum for precipitation) of the climate model is conserved and reproduced by the
disaggregated values.
For temperature, the resulting hourly temperature is further scaled between the provided minimum
and maximum. The scaling is performed in a way that the daily mean value is preserved with an
accuracy of four decimals. Relative humidity is limited to 100%, considering the preservation of the
daily mean value.
Large selected DOY windows increase the statistical population, but on the other sight might distort
climatic characteristics with a strong seasonal course such as shortwave radiation values for the actual
DOY. Therefore, we preprocessed hourly potential (cloud free) solar radiation for each DOY globally at
0.5° spatial resolution. This data is used as upper bound to limit the resulting hourly values for the
corresponding DOY, while the daily mean value is preserved.
In a final step, the hourly values are aggregated to the temporal resolution as set by the user.

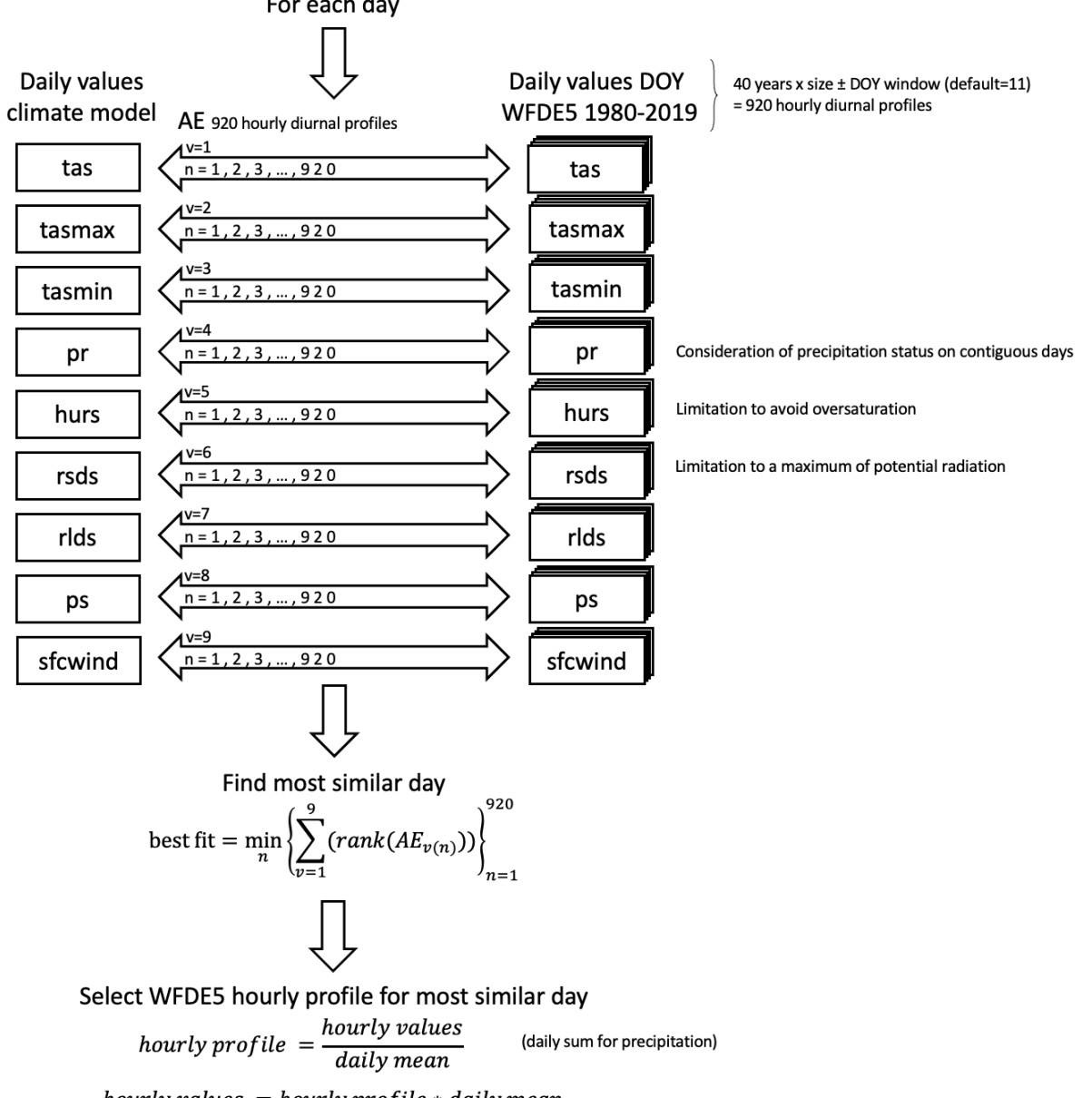

Figure 1: Procedure to identify the most similar meteorological day in the population of WFDE5 reference data for the default DOY window of ± 11 days around the actual DOY. Daily values refer to daily sum for precipitation and daily mean values for all other variables.

In rare cases, precipitation cannot be distributed, due to no precipitation in the reference data. This can happen in dry deserts, where 40 years of WFDE5 data show no precipitation record within the range of the moving DOY window (Supplementary Fig. S1 shows a map where this is the case). To handle this exception, several options are implemented. First, the DOY window is automatically expanded to +-50 days around the actual DOY in order to increase the statistical population and thus the probability to include a precipitation event. If still no precipitation event is found in the reference, a linear regression between the precipitation amount and the precipitation duration is performed for the specific location across the entire available data spectrum. The linear regression determines the usual duration of the selected precipitation event. Subsequently, an hour is randomly selected for the start of the precipitation event. A goal of Teddy was to consider the physical consistency of inter-variable relationships. Precipitation generally affects other climate variables (e.g. humidity, radiation,

temperature, etc.; Meredith et al., 2021). During night, physical interdependencies between precipitation and other variables are generally lower, because radiation is not affected and less energy is available to affect other variables. This might have an effect for impact models, because, as an example, evapotranspiration might be unrealistically high if precipitation occurs at the same time with full solar irradiation during noon. In order to reduce possible inconsistencies with other variables that could lead to implications in impact models, the precipitation is only distributed to hours at nighttime. Alternatively, we implemented the option for the user to write Not a Number (NaN) values instead.

Drizzle precipitation (values below 1 mm day$^{-1}$) is also disaggregated to sub-daily values in order to ensure mass and energy conservation. If no historical precipitation event is found for this case, precipitation noise is again randomly distributed to an hour at nighttime. If no hour without radiation occurs (e.g. high latitudes in northern summer), the precipitation is distributed to local midnight.

The calculation procedure can be performed either for universal time (UT) or for local solar time (LST). The latter divides the world into equal time zones of 15° with the central time zone (+-7.5°) at Greenwich.

## 4. Results

In a first step, Teddy is applied for 30 globally distributed samples (Fig. 2) for the year 2010. To be able to validate the results, we perform a cross-validation. Therefore, WFDE5 data for 2010 aggregated to daily values serve as an input for Teddy. The same year is excluded from the statistical population during the cross-validation. As a result, it can be tested how well WFDE5 hourly values for the year 2010 are reproduced with the statistical population of the other 39 years. The 30 samples are chosen to represent globally relevant agricultural production regions in different climate zones (Fig. 2). To evaluate the sensitivity of the different DOY window sizes, we run the cross-validation with different DOY window sizes, ranging from 1 to 25, in steps of two, including the option to disable the DOY window (DOY window size = 0). In order to additionally validate the performance for extreme events, we perform a second cross-validation for all available 40 years (1980-2019) with DOY window sizes of 11 for sample location 29, located in Southern Germany.

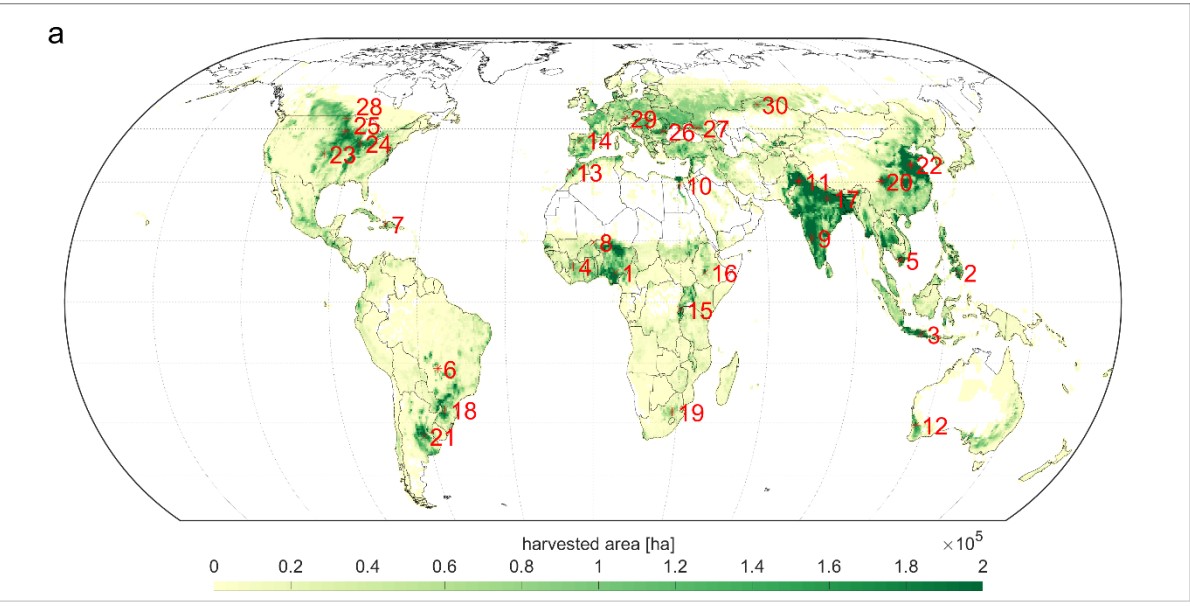

209

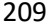

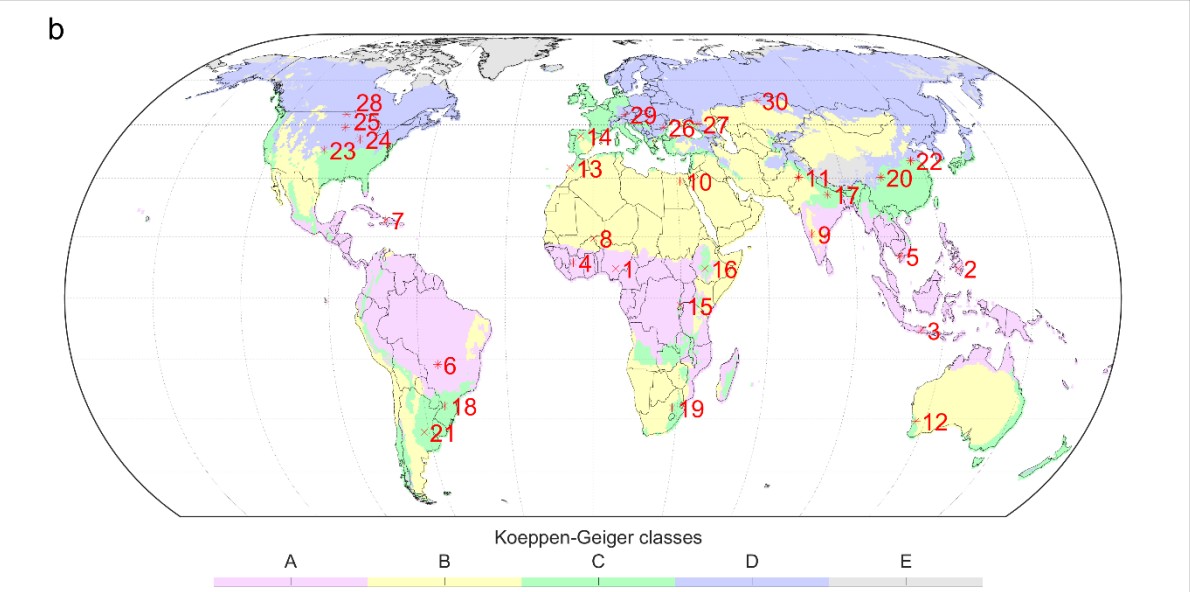

210

Figure 2: Distribution of 30 global samples used for the cross-validation on (a) annual total harvested area of rainfed and irrigated crops in hectare per pixel at a 30 arc-minute grid (Portmann et al., 2010) and (b) for Koeppen-Geiger climate zones calculated for 1980-2019 WFDE5 temperature and precipitation values (Beck et al., 2018). Samples are ordered by climate zone affiliation and their distance to the equator.

4.1 Validation

As an example, for sample location 16 in Ethiopia, Fig. 3 shows the results of the temporal disaggregation series for the cross-validation for a 10-day time series in 2010 in comparison with the daily climate input and the original hourly WFDE5 data. The hourly courses show high correlations for the randomly selected time series for all variables except for precipitation (Fig. 3 and scatterplots in Fig. 4 for the entire year; Supplementary Fig. S2 and S3 alternatively show sample location 22 in China).

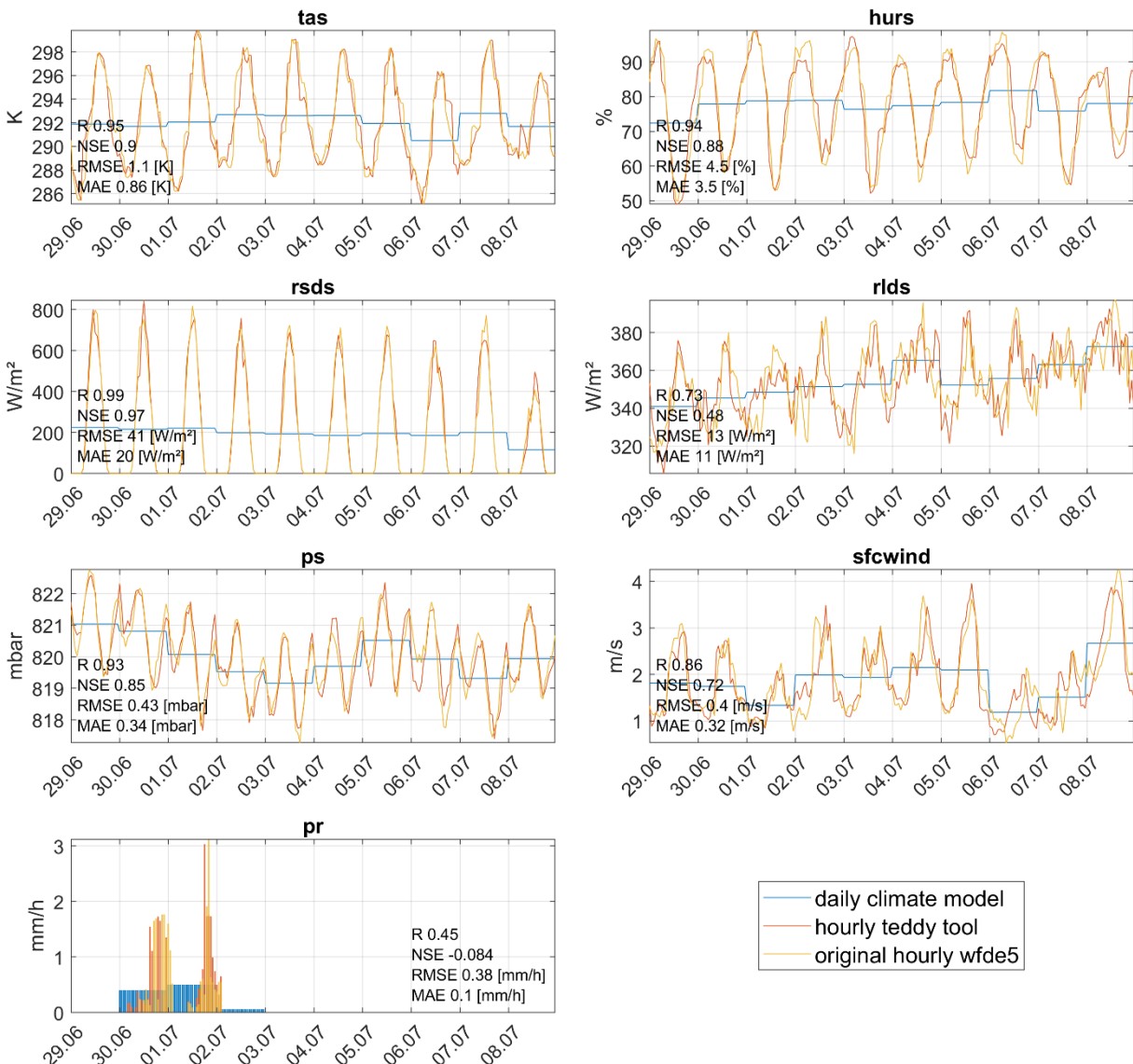

222

Figure 3: Time-series for all variables comparing daily climate model data, disaggregated hourly results of Teddy from the performed cross-validation and the original hourly WFDE5 data, shown for sample location 16 in Ethiopia with a DOY window size of 7 for the 10-day period 29.06. – 08.07.2010. The Pearson correlation coefficient (R), the Nash-Sutcliffe model efficiency coefficient (NSE), the root mean squared error (RMSE) and the mean absolute error (MAE) are displayed for the shown time period for each variable.

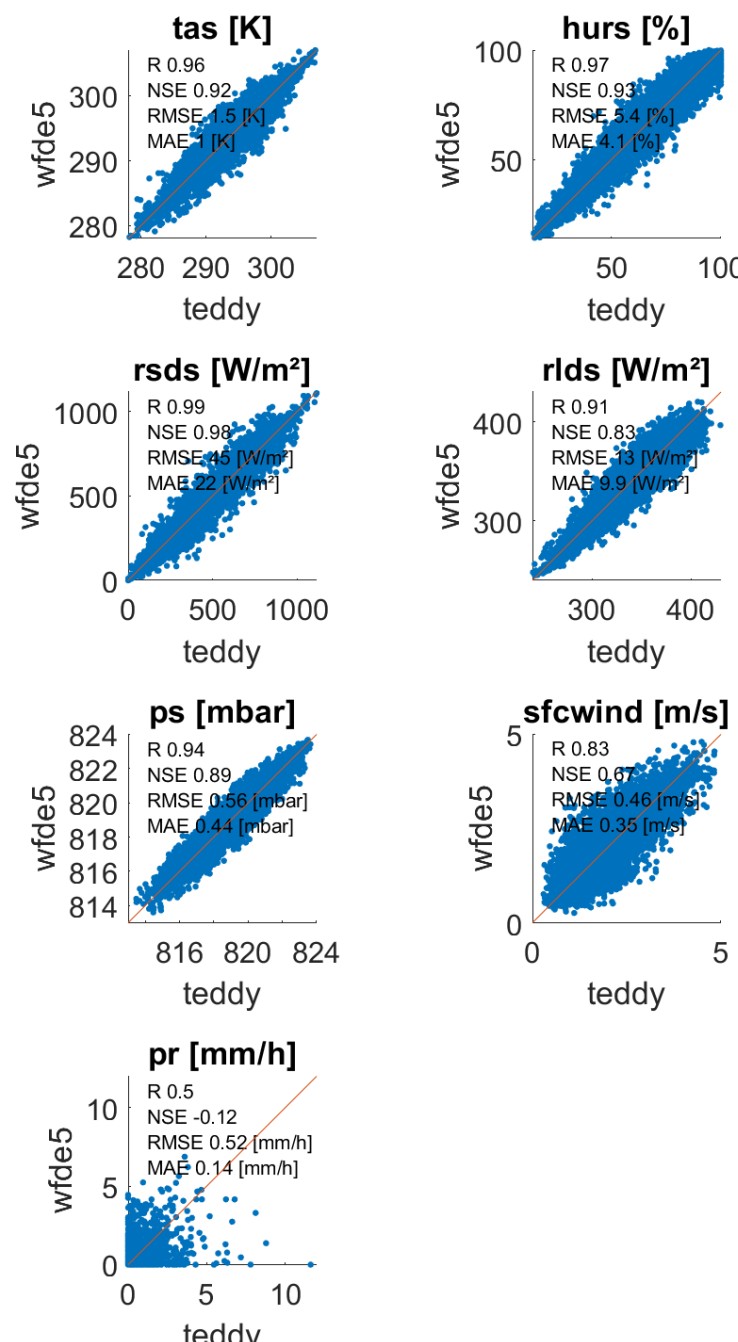

229

Figure 4: Hourly values for the year 2010 between disaggregated values generated by the Teddy-Tool and the original WFDE5 data used for the cross-validation, exemplarily for sample location 16 in Ethiopia with a DOY window size of 7. The Pearson correlation coefficient (R), the Nash-Sutcliffe model efficiency coefficient (NSE), the root mean squared error (RMSE) and the mean absolute error (MAE) are displayed for each variable.

### 4.2 Sensitivity analysis DOY window size

The sensitivity analysis averaged over all 30 samples shows that the Pearson correlation coefficient of hourly values for the year 2010 show high correlations for all variables (r>0.9), except wind speed (r>0.7) and precipitation (r>0.4), which are generally more difficult to disaggregate (Fig. 5; Supplementary Fig. S4 additionally shows the Nash-Sutcliffe model efficiency coefficient). The selected DOY window size has an effect on the quality of the results. While no DOY window (size=0) results in

the lowest correlation coefficient across all variables, the DOY window size does significantly affect the
correlation for precipitation and wind speed (Fig. 5).

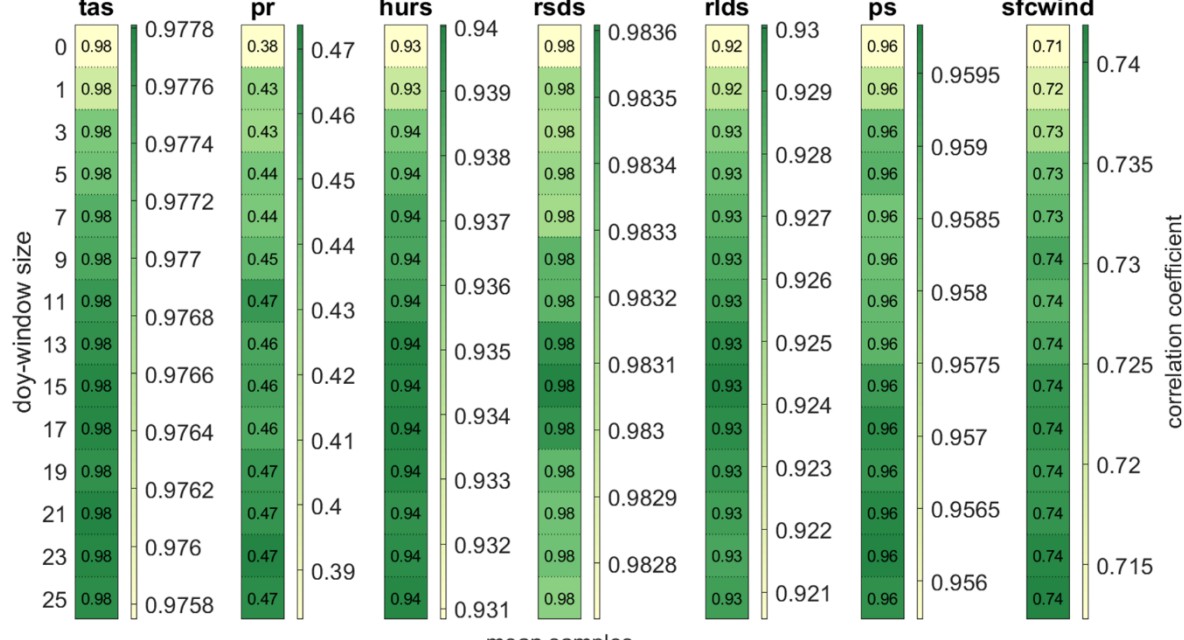


Figure 5: Pearson correlation coefficient between disaggregated hourly values generated by the Teddy-
Tool and the original WFDE5 data used for the cross-validation for different DOY window sizes
averaged over all 30 samples for the year 2010 for all variables. The scaling of the colorbar differs
between variables.
For precipitation, the impact of the DOY window size on the correlation varies between regions. Larger
DOY windows are mainly beneficial for precipitation in arid regions, while showing lower increases in
correlation in regions with pronounced seasons (Fig. 6). The results also show that the correlation for
precipitation is generally larger in tropical regions than in continental regions.

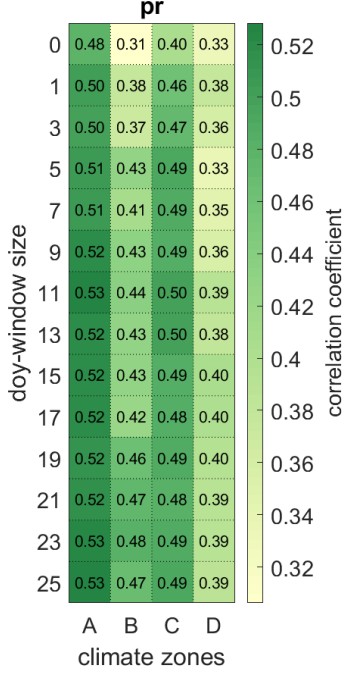


Figure 6: Pearson correlation coefficient between disaggregated hourly values generated by the Teddy-Tool and the original WFDE5 data used for the cross-validation for different DOY window sizes averaged over the samples for each Koeppen-Geiger climate zone (A=tropical, B=arid, C=temperate, D=continental).

While hourly precipitation can be best reproduced for winter seasons in continental and arid regions, winter seasons show the lowest correlation for temperate regions. Tropical regions only show relatively low variations over the year, independently from the selected DOY window size (Fig. 7). Especially in arid regions, the length of the DOY window size affects the results differently in different seasons. Here, larger DOY windows decrease the correlation during the rainy season (winter and spring), while correlation is increased during the dry season (summer and autumn).

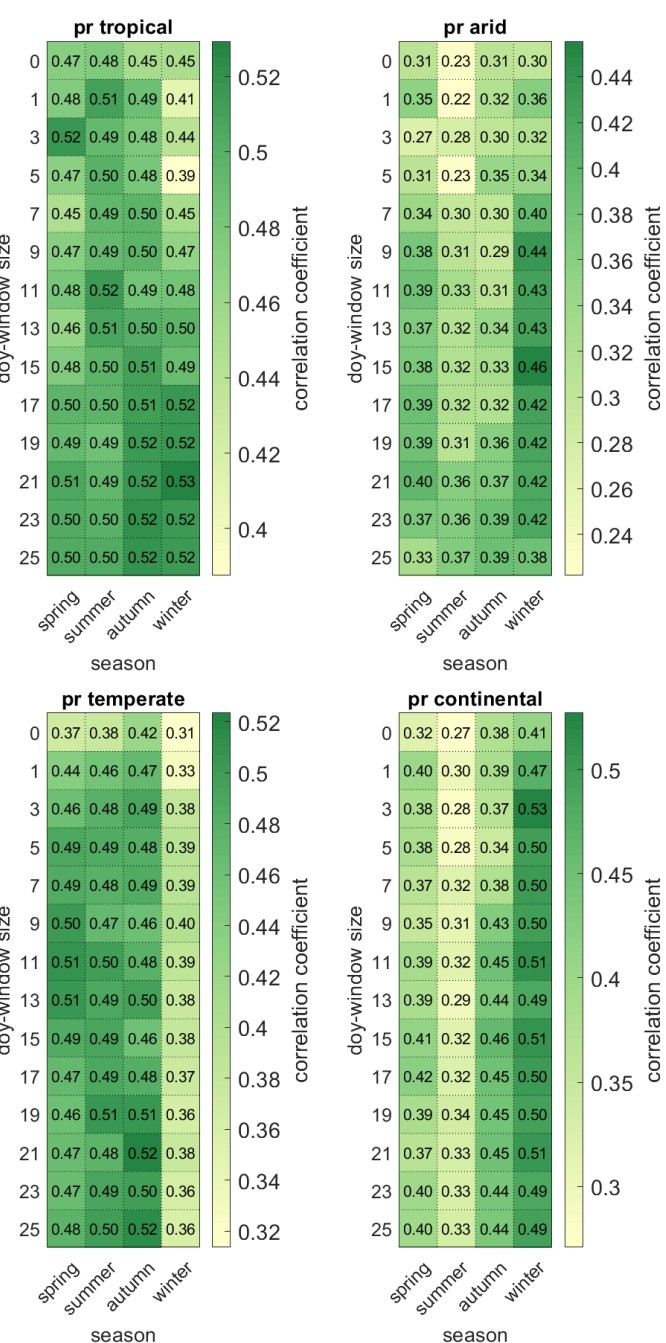

Figure 7: Pearson correlation coefficient between disaggregated hourly values generated by the Teddy-
Tool and the original WFDE5 data used for the cross-validation for different DOY window sizes
averaged over the samples for the four seasons (Northern hemisphere: spring=MAM, summer=JJA,
autumn=SON, winter=DJF; Southern hemisphere: spring=SON, summer=DJF, autumn= MAM,
winter=JJA). The heatmap is averaged over the samples for each Koeppen-Geiger climate zone
(A=tropical, B=arid, C=temperate, D=continental).
Furthermore, we evaluate the sensitivity of the DOY window size to the reproduction of temporal
autocorrelation (Fig. 8). Therefore, the autocorrelation over lag times between one and 24 hours is
calculated for precipitation and wind speed. Autocorrelation refers to the similarity of a time series to
a lag duration shifted version of the same time series. This allows sub-daily patterns and inter-hour
connectivity to be statistically captured and validated in time series of precipitation and wind speed.
In addition, we also check the reproduction of wet hours (precipitation above 0.1 mm h$^{-1}$) in 2010 and
the number of hours with low wind speeds (sfcwind < 2.5 m s$^{-1}$) referring to the typical cut-in wind
speed of wind turbines.
Here, we find that short DOY window sizes below 5 days are not beneficial to all statistics. The
autocorrelation of precipitation (wind speed) is reproduced more accurately with window sizes of 9
days or longer. The number of wet hours is better recreated with window sizes above 15 days. For
hours with low wind speed, a minor improvement is found above 9 days.

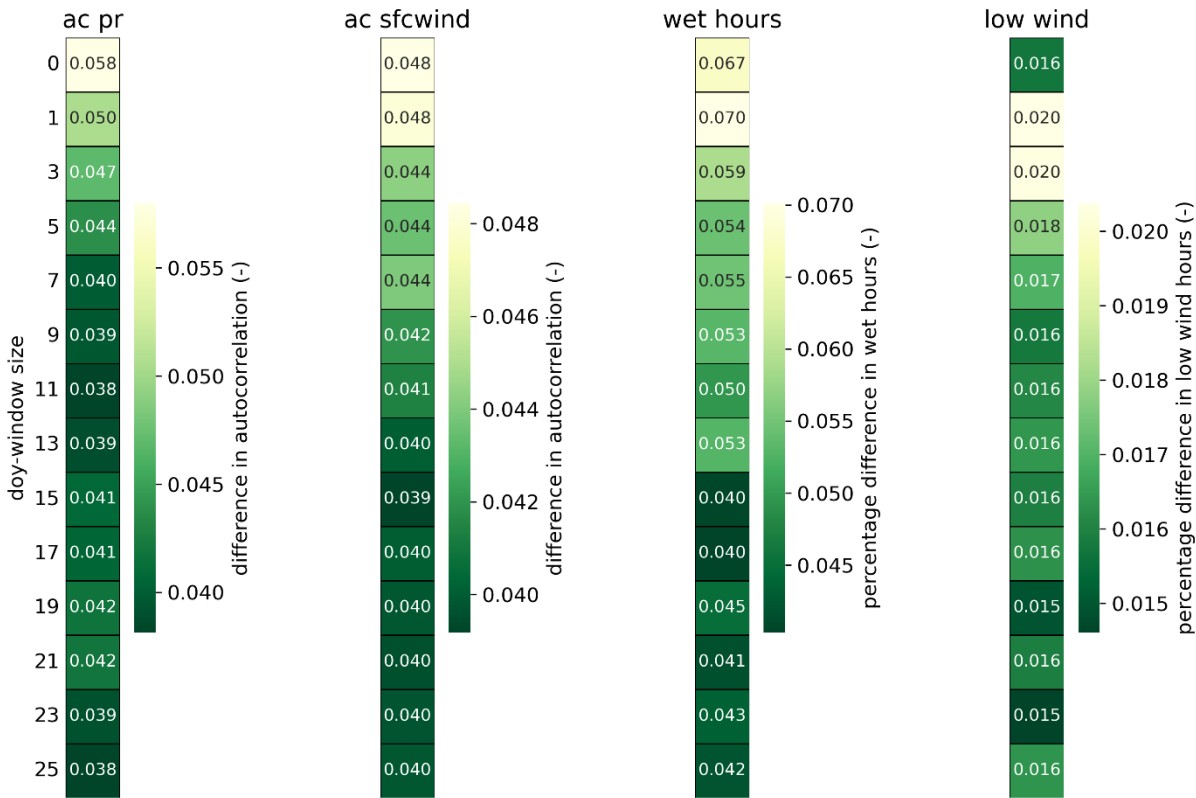


Figure 8: Extended validation statistics for the sensitivity analysis of the DOY window size for the year
2010. The difference in autocorrelation refers to the average over all 30 samples and lag durations
between one and 24 hours. Wet hours are defined as precipitation intensities above 0.1 mm h$^{-1}$ and
low wind speeds refer to hours with sfcwind < 2.5 m s$^{-1}$.
4.3 Multi-year evaluation
The previous validation has assessed the disaggregation performance for all sample locations for the
year 2010 and different DOY window sizes. For the analysis of the whole time period 1980 – 2019, we
evaluate each year of the 40-year timeseries for sample location 29 and a window size of 11 days.
Figure 9 and Supplementary Fig. S5 show the correlation coefficient and mean absolute error,
respectively, for each year to assess the interannual variability of disaggregation performance. For tas,
hurs, rsds, rlds, and ps the performance shows only very minor differences, whereas sfcwind and pr
show a higher degree of interannual fluctuations.

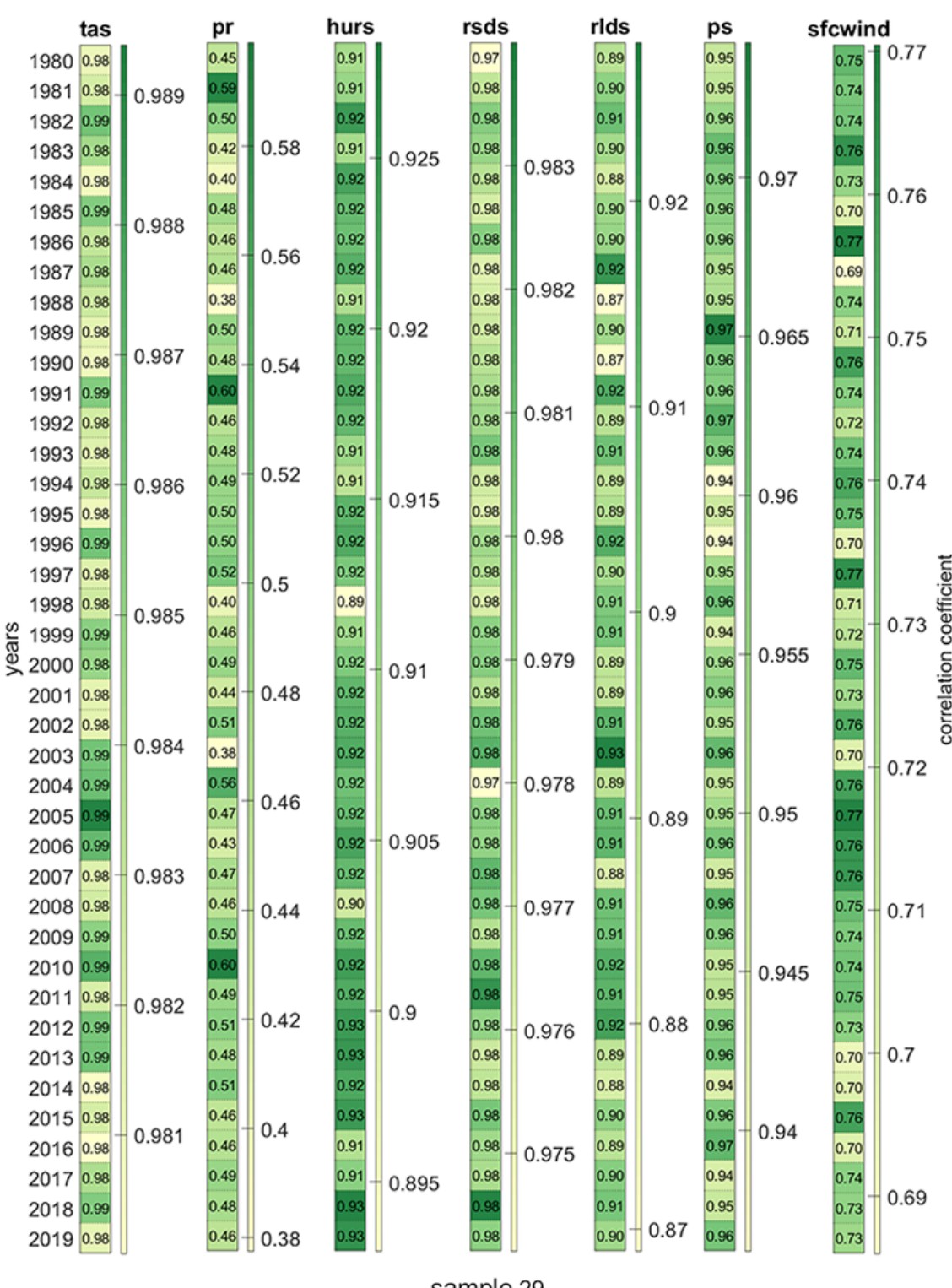

sample 29

Figure 9: Pearson correlation coefficient between disaggregated hourly values generated by the Teddy-Tool and the original WFDE5 data used for the cross-validation for each year from 1980 to 2019 for sample location 29 and a DOY window size of 11 days. The scaling of the colorbar differs between variables.

4.4 Evaluation of precipitation: Wet proportions and intensities

For the further evaluation of precipitation characteristics, we additionally assess the disaggregated
timeseries over the whole period 1980 − 2019 for sample location 29. In order to evaluate the
reproduction of wet/dry proportions, the monthly cycle of wet hours is provided (Fig. 10). Wet hours
above 0.1 mm h$^{-1}$ are recreated by the Teddy-Tool with minor differences for the median over 40 years
(Fig. 10). The error measures are calculated for every year separately amounting to a mean absolute
error of 13.02 h equaling 7.8 %.
For the evaluation of the range of precipitation intensities, Fig. 11 shows intensities above 1 mm h$^{-1}$
plotted against its percentage of exceedance for sub-daily durations. We find that the disaggregated
precipitation intensities match the original data except for extreme precipitation.

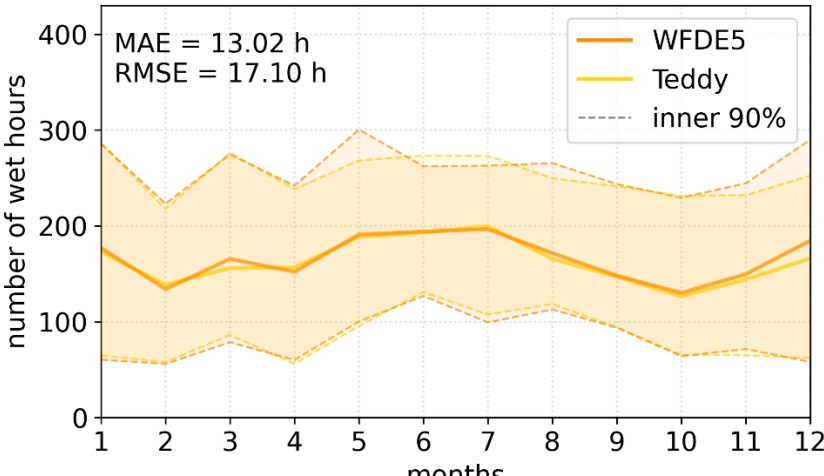


Figure 10: Number of wet hours per month for sample location 29 in Germany. Solid lines show the
median over 40 years, where the dashed lines denote the inner 90% of the 40-year period. MAE and
RMSE are calculated separately for every year and averaged over 40 years.


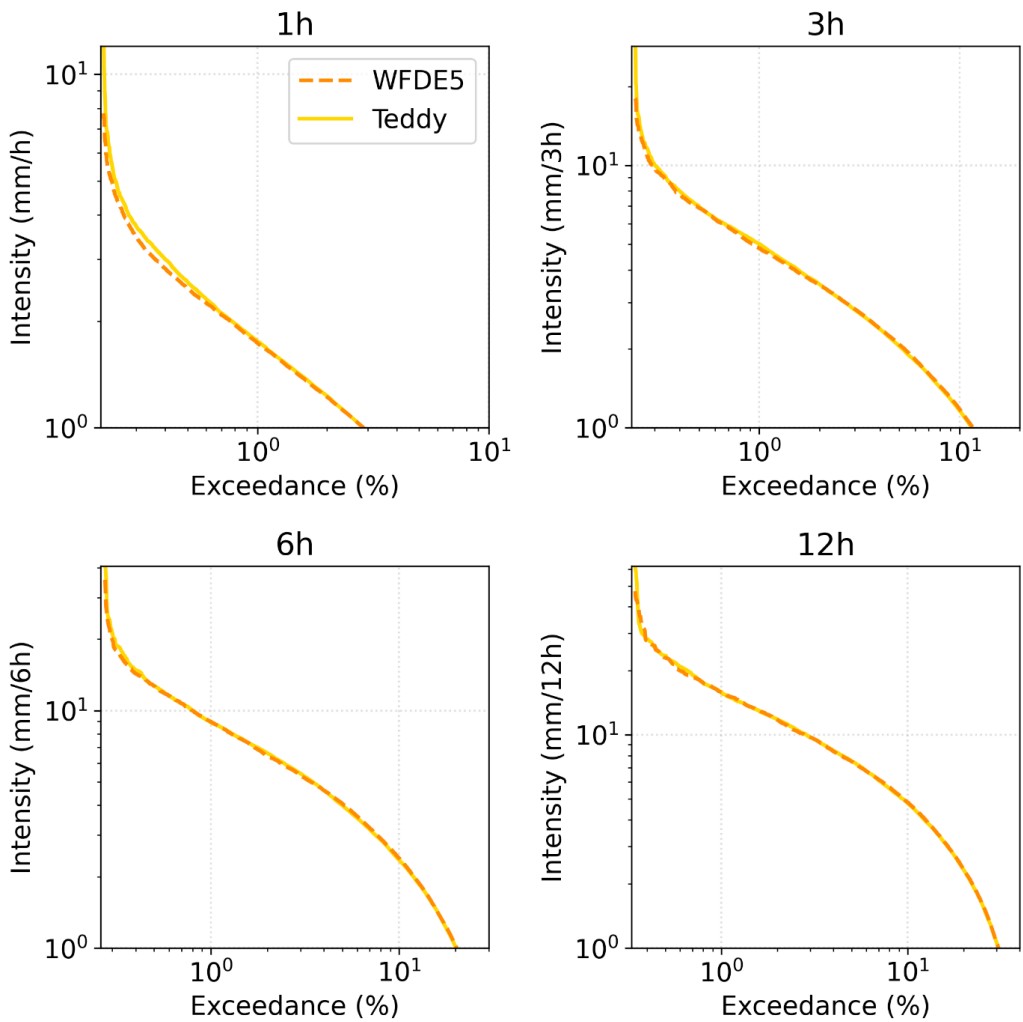


Figure 11: Exceedance probability of precipitation intensities for sub-daily durations for sample
location 29 in Germany.

4.5 Evaluation of precipitation extremes

As the ISIMIP data base is used for future impact modelling and historical attribution science (Mengel
et al., 2021), extremes are of major interest for the community. The ability of global climate models to
simulate sub-daily extremes is limited and depends on the variable of interest and the spatio-temporal
conditions of the extreme and the respective model setup (Wehner et al., 2021; Kumar et al., 2015;
Wang and Clow, 2020). However, in this validation, we evaluate how the Teddy-Tool is able to preserve
the statistics of sub-daily extreme values. Therefore, we select precipitation as variable of interest.
Figure 12 shows the reproduction of sub-daily precipitation extremes for 1980 – 2019 for sample
location 29 in southern Germany, where Teddy is run with a DOY window size of 11 days. The 40 annual
maxima are extracted from the original and the disaggregated data. Additionally, the Generalized
Extreme Value (GEV) distribution is fitted to these empirical data. GEV parameters are estimated via
Maximum Likelihood Estimation (Coles, 2001), where the goodness-of-fit is assessed with the
Anderson-Darling test at 95% significance level (Stephens, 1986). Thereby, 95% confidence intervals
are generated applying a bootstrap procedure with 1000 iterations to account for extreme value
statistical uncertainties. We find that the Teddy-Tool leads to an overestimation of annual maximum
precipitation. For the hourly duration, the differences are large with the confidence intervals of the

GEV hardly overlapping. For the longer durations, Teddy values approach the original data, with noticeable differences only for the rare events with return periods above 5 years.

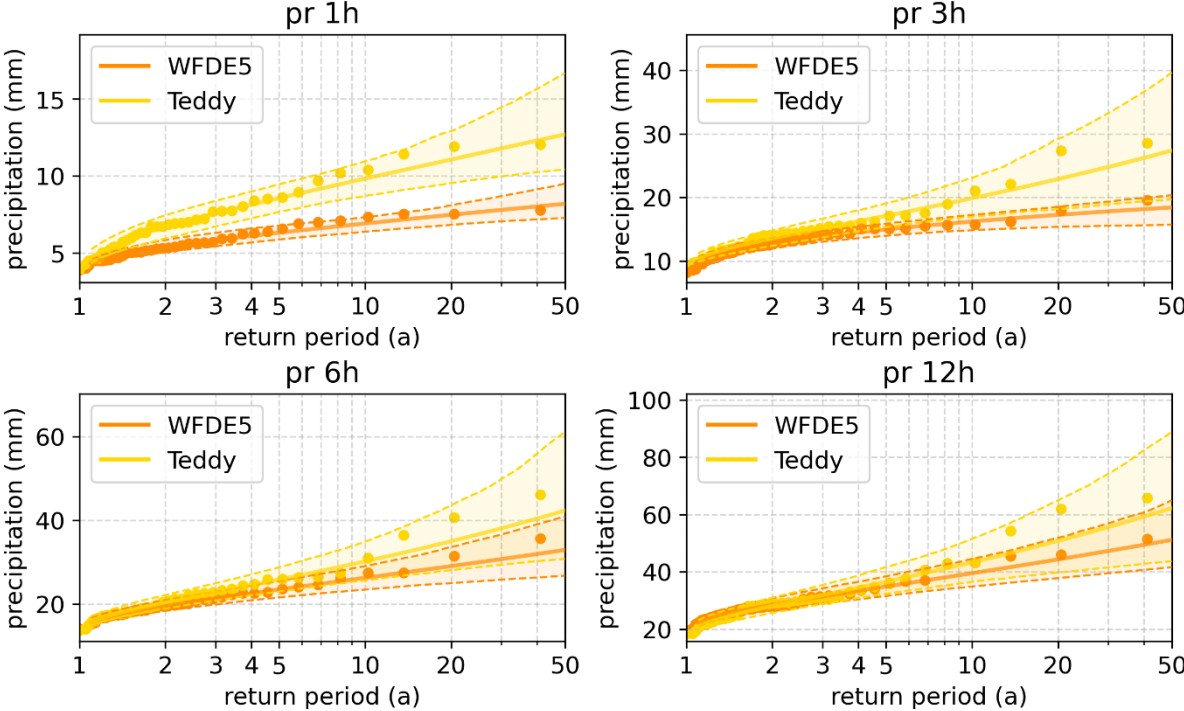

Figure 12: Extreme value statistical evaluation of sub-daily precipitation for sample location 29 in Germany. The annual maxima of the WFDE5 and Teddy are shown as dots. Additionally, GEV fits (lines) with 95% confidence intervals (transparent areas and dashed lines) account for uncertainties. The Teddy-Tool is run with a DOY window size of 11 days.

## 5. Discussion and Outlook

The Teddy-Tool allows for temporal disaggregation of daily climate model data. The disaggregation is based on location and time specific empirical relationships between variables. The approach is well suitable for all tested variables and results in very high correlations (>0.9), except for precipitation (>0.5) and wind speed (>0.75). We refer the worse performance for precipitation and wind speed to the high intra-day variability for these variables (Watters et al., 2021). Other variables are governed by a stronger diurnal cycle (Dai and Trenberth, 2004), which is easier to disaggregate based on empirical diurnal profiles.

Compared to other approaches, the advantage of the Teddy-Tool is that no other input data is required rather than the daily climate model data. The Teddy-Tool is relatively simple to apply, considers specific local and seasonal features of the diurnal course of different climate variables, and preserves the physical consistency of inter-variable relationships. Mass and energy are conserved and mean daily values of the climate model are reproduced any time.

The spatial and temporal resolution of the results is determined by the provided temporal and spatial resolution of the chosen reference data (WFDE5 used here). Longer available reanalysis time periods extend the statistical population for identifying the most similar weather conditions in the past and thus could improve the results. Generally, also other reference data could be used, that provides higher temporal or spatial resolution for a specific region.

The DOY window to find the most similar historical weather situations can be chosen in different sizes.
For most of the variables, we found small effects of time window adjustments, except for precipitation
and wind speed. The evaluation of different DOY window sizes reveals that a DOY window size of 11
can generally be recommended across all variables. Larger DOY windows should be avoided mainly in
arid regions, while shorter DOY windows generally lead to poorer representations of autocorrelation
and extreme events.
One limitation of the Teddy-Tool is the representation of extreme events, mainly for precipitation,
which is generally the most difficult variable for temporal disaggregation. We found that hourly
precipitation extremes are overestimated. For heavy daily precipitation events, Teddy distributes the
24h-sums either correctly, too evenly or on too few hours. When distributing on too few hours,
extreme hourly intensities evolve, which may have never occurred or may even be physically
implausible. For temporal disaggregation of extreme precipitation, we recommend dynamical
downscaling via high-resolution climate models (Poschlod, 2021; Poschlod et al., 2021; Zabel et al.,
2012; Zabel and Mauser, 2013).
Another limitation of the approach is the reproduction of the inter-day connectivity within the
disaggregated time series. When two diurnal profiles are chosen for the disaggregation of adjacent
days, which show dissimilar courses in the time steps at the change of the day, abrupt value jumps
might occur in the disaggregation. This can be seen in Fig. 3 for rlds from July 4$^{th}$ to July 5$^{th}$. To illustrate
this issue, a disaggregation time series from another location is provided in Supplementary Fig. S2. This
limitation does also apply for the Method of Fragments applied on precipitation (Li et al., 2018).
Similarly to Li et al. (2018), we also consider the precipitation state of the previous and following day
to improve inter-day connectivity. Without this additional consideration, overnight precipitation
events would often be 'cut off' in the disaggregation. For the remaining abrupt jumps in the
disaggregated time series, we refrain from post-processing with subsequent smoothing, as we want to
preserve both mass and energy and the empirical diurnal profiles.
For the disaggregation of future climate projections using of the Teddy-Tool, we have the following
remarks: As the Teddy-Tool derives the relationships between sub-daily and daily values empirically
based on reanalysis data, future diurnal profiles, which are outside the historical range of diurnal
profiles, might possibly be not fully reproduced. However, this limitation is common for statistical
approaches, which are to be calibrated on historical data (Papalexiou et al., 2018). Nevertheless, due
to energy and mass conservation, climate trends in the daily climate signal are fully preserved. Hence,
applying Teddy for temporal disaggregation under climate change holds under the assumption that we
select the most similar meteorological day of the historical data and that this diurnal profile is
representative for future climatic conditions. However, this assumption might apply to a different
degree for different variables. We expect non-stationarity for the diurnal profiles due to changing
weather patterns, shifts in rainfall generating processes, and shifts in the seasonality, mainly for
precipitation and wind. The daily course of other variables, such as solar radiation and temperature
might generally be less affected by a warmer climate. Furthermore, global climate models at coarse
resolutions generally do not represent all processes to fully reproduce intra-day variability. Teddy
applies the diurnal profiles and intra-day variability from the WFDE5 data, which are bias-adjusted
ERA5 reanalysis data that implicitly consider finer scale effects than coarse-resolution global climate
models (Cucchi et al., 2020). Thus, the disaggregation process in Teddy is consistent with the bias
adjustment in ISIMIP3.
Another limitation of the methodology could occur in the case of strong climate change signals. In case
of high warming in end-of-century projections, the number of sampled historical days might decrease
if the same historical day is sampled repeatedly. This could lead to reductions in diversity of the diurnal
profile. Hence, Teddy allows to monitor the number of unique analogue days per year. An additional
analysis for SSP3-7.0 using the GFDL-ESM4 climate model shows that the number of unique analogue
climate days are declining, as expected, but still the diversity of chosen days is above 300 unique days
at the end of the century for a chosen moving-window size of +-11 days (Supplementary Fig. S6). A
smaller size of the moving window prevents that the same analogue day is chosen over a longer time
period. This will increase the diversity of diurnal profiles at the expense of similarity. Even if diurnal
profiles are derived from the same analogue day repeatedly, the disaggregated diurnal courses, e.g.
for temperature, will show variations (different offset and different amplitude) due to conservation of
daily mean energy and mass. From a broader perspective, it is also not clear whether the uncertainties
resulting from this limitation are larger than the uncertainties within the climate model projections
until the end of the century. Furthermore, in the long term, the basic population for finding analogue
climates will continuously increase, since WFDE5 data, which are based on ERA5, are continuously
updated. We note that Teddy could be also employed to disaggregate future daily climate projections
based on hourly future climate projections as reference.
Further possible developments could include improvements for the reproduction of the inter-day
connectivity. Despite the consideration of precipitation classes, still abrupt value jumps over day
changes are possible. A future introduction of temperature classes and surface pressure classes in
addition to the precipitation classes could help to reduce this effect. Depending on the location of
interest, also including climate modes or weather patterns for the choice of the most similar
meteorological day could positively affect the performance. Furthermore, depending on the
application, it could be reasonable not to screen for the most similar meteorological day, but for the
most similar succession of multiple days. This would as a consequence improve the inter-day
connectivity as less different profiles are selected.
Other optional future developments could include the separation of direct and diffuse radiation, which
is also a required information for some impact models which is currently not provided by ISIMIP.
However, we would make further development with more options dependent on the community's
adoption of the current executable tool.
**Code availability**
The source code of the Teddy-Tool (v1.1) and a parallelized version of the Teddy-Tool (v1.1p), including
a precompiled executable file for Windows, preprocessed data, results of the cross-validation and
exemplary results for SSP 585 (2015 – 2100) and the UKESM1-0-L climate model for 30 samples are
provided via Zenodo (https://doi.org/10.5281/zenodo.8124111).
**Author contribution**
FZ: Conceptualization, Software, Methodology, Validation, Formal analysis, Resources, Data curation,
Writing - original draft, Visualization
BP: Methodology, Validation, Formal analysis, Writing - original draft, Visualization
**Competing interests**
The contact author has declared that none of the authors has any competing interests.

## Acknowledgements

We acknowledge the methodological discussion with Stefan Lange from the Potsdam Institute of
Climate Impact Research (PIK).

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
