# Peer review of "The Teddy-Tool v1.1: temporal disaggregation of daily climate model data for climate impact analysis"

_Geoscientific Model Development, 2023_

## Author Comment (AC4)

[revised manuscript text omitted]

For each day

Daily values climate model

Daily values DOY WFDE5 1980-2019

years x size ± DOY window (default=11) = 920 hourly diurnal profiles

AE 920 hourly diurnal profiles

| | |
|---|---|
| tas | $v=1$ $n = 1, 2, 3, …, 9\,2\,0$ |
| tasmax | $v=2$ $n = 1, 2, 3, …, 9\,2\,0$ |
| tasmin | $v=3$ $n = 1, 2, 3, …, 9\,2\,0$ |
| pr | $v=4$ $n = 1, 2, 3, …, 9\,2\,0$ |
| hurs | $v=5$ $n = 1, 2, 3, …, 9\,2\,0$ |
| rsds | $v=6$ $n = 1, 2, 3, …, 9\,2\,0$ |
| rlds | $v=7$ $n = 1, 2, 3, …, 9\,2\,0$ |
| ps | $v=8$ $n = 1, 2, 3, …, 9\,2\,0$ |
| sfcwind | $v=9$ $n = 1, 2, 3, …, 9\,2\,0$ |

tas · tasmax · tasmin · pr (Consideration of precipitation on consecutive days) · hurs (Limitation to avoid oversaturation) · rsds (Limitation to a maximum of potential radiation) · rlds · ps · sfcwind

Find most similar day

$$\text{best fit} = \min_n \left\{ \sum_{v=1}^{9} (rank(AE_{v(n)})) \right\}_{n=1}^{920}$$

Select WFDE5 hourly profile for most similar day

$$\text{hourly profile} = \frac{\text{hourly values}}{\text{daily mean}} \quad \text{(daily sum for precipitation)}$$

[revised manuscript text omitted]

---

## Author Response (AR1)

Dear Reviewer #1,

We thank you for your time to review our submitted paper and we greatly acknowledge your valuable comments, which help us to improve our paper. In the following, we reply to your general and specific comments. Thereby, our reply is placed right below your comment and marked bold.

**Reviewer #1:**

The authors present a tool for temporal disaggregation of daily climate model outputs to a sub-daily hourly time step. The tool covers the standard suite of meteorological driving variables required for land surface modelling, for example driving a distribution hydrological model or offline land surface scheme. Unlike many existing off-the-shelf approaches, Teddy-Tool is an empirical approach that searches for the best diurnal cycle analogue, matched based on a time window of daily outputs, using the globally available bias-corrected hourly reanalysis WFDE5 data (1980–2019) as the analogue pool. The approach is evaluated using a "perfect observation" approach where a single WFDE5 year is withheld from the analogue pool, hourly data are aggregated to daily, and performance of the method is assessed.

**Reply:** This is well summarized.

The tool is written in MATLAB and a full set of example data and source code are provided in a public repository.

Overall, my impression of the paper is positive. It fills a needed gap using an approach that is simple, provides physically-consistent inter-variable relationships, and makes sense for near-term future projections or seasonal to decadal climate predictions.

**Reply:** Thank you for your positive feedback and your note for the need and gap-filling of our approach, considering physically consistent inter-variable relationships.

My main concern is that the approach may lead to some pathological behavior when faced with strong climate change signals, for example end-of-century projections for moderate to high greenhouse gas emissions scenarios. Because the approach is built around the idea that the "diurnal profile of all variables is taken from the same, most similar meteorological day of the historical reanalysis dataset" it is conceivable that, under strong climate change, the same analogue day (or a very small number of analogue days, for example the hottest historical days for a given part of the annual cycle) will be sampled repeatedly, thus leading to a reduction in diversity of the diurnal profiles. Similar issues can exist, albeit on different time scales (e.g., monthly to daily disaggregation) for some downscaling algorithms like BCSD (Raff et al., 2009).

**Reply:** Regarding your concern of high warming end-of-century projections, we basically agree that this is a methodological constraint. We are aware of this limitation and already discuss this issue in detail (see line 292ff). To address your concerns, we add a counter in the Teddy-Tool, which diagnoses the number of different historic diurnal profiles applied per year as shown in Fig. 1. So, the user is able to monitor this methodological constraint. In addition, we would like to put into perspective that a smaller size of the moving window (which can be set by the user) prevents that the same analogue day is chosen over a longer time period. This

will increase the diversity of diurnal profiles at the expense of similarity. Further, since mass and energy are conserved within the disaggregation approach, the diurnal course, e.g. for temperature, might show variations (different offset and different amplitude) despite the diurnal profile are derived from the same analogue day.

From a broader perspective, it is also not clear whether the uncertainties resulting from this limitation are larger than the uncertainties within the climate model projections until the end of the century. Furthermore, in the long term, the basic population for finding analogue climates will continuously increase, since WFDE5 data, which are based on ERA5, are continuously updated.

We added these aspects to the discussion.

This is noted as a possibility, but the degree to which it is a problem is not tested. How many distinct analogue days are sampled in climate projections from SSP3-7.0 for the end of the 21st century? It would be nice to see this being evaluated, especially given that the stated goal is the disaggregation of climate model outputs. Similarly, how does the method compare with existing parametric approaches? Is there value added by this method? Is the performance comparable to Bennet et al. (2020) or Forster et al. (2016)?

**Reply:** Thanks a lot for this suggestion. It is a great idea to evaluate the number of analogue climate days. As suggested, we tested this in an additional analysis for SSP3-7.0 using the GFDL-ESM4 climate model (see Fig. 1). The result for 607 samples, distributed over the entire USA (including Alaska and Hawaii), shows that the number of unique analogue climate days are declining, as expected, but still the diversity of chosen days is above 300 at the end of the century for a chosen moving-window size of 11 days. We suppose that this is far away from a critical range. We included Fig. 1 to the results and refer to it in the discussion.

[Figure]

*Figure 1:* Number of unique analogue climate days per year for the GFDL-ESM4 climate model under SSP 3-7.0 for the years 2015-2100 and a selected moving window size of 11 days, showing the mean (blue) and median (orange), as well as the range between the 5[th] and 95[th] percentile for a number of 607 samples in the USA.

Regarding the suggestion for comparing our method with other approaches, please see the next reply.

Noted above is the similarity of the method to existing empirical disaggregation schemes, like the one in BCSD, but there are also others like the method of fragments (Li et al., 2018) or the historical analog (Chen, 2016) approach. An expanded literature review that covers existing analogue or k-nearest neighbour approaches in climatology would be useful.

**Reply:** The intention of this publication is not a model comparison, although we agree that this would be really interesting for a follow-up publication. In a follow-up publication, also the performance in terms of computational demand and computing time can be compared. In this context, we would like to mention that in the meantime of this review process, we parallelized the Teddy-Tool and thus significantly decreased the computing time. The parallelized version of Teddy is uploaded to Zenodo and provided Open Source under CC-BY license.

Thanks a lot for the suggested additional publications. We already refer to several different and similar disaggregation approaches (see line 46-60) and already refer to Li et al. (2018) in line 121 and Förster et al. (2016) in line 49 and 57. We complemented our description with Chen (2016), and Bennet et al. (2020) as you suggested.

One more minor concern is that the software is written in MATLAB, which is a proprietary, commercial software package. Will the method run with a free and open source alternative like Octave?

**Reply:** Matlab is platform independent and a precompiled version is provided, which can be used without any license. Therefore, a redistributional package of Matlab must be installed before running the executable file. Since the code is provided open source, anyone can translate the code into other languages as preferred.

We included this information to the data availability statement, also mentioning the availability of a parallelized version.

Dear Reviewer #2,

first, we want to thank you a lot for your time to review our manuscript and for providing such helpful comments! In the following, we reply to your general and specific comments. Thereby, our reply is placed right below your comment and marked bold.

**Reviewer #2:**

This manuscript, titled "The Teddy-1 Tool v1.0: temporal disaggregation of daily climate model data for climate impact analysis", presented a method to sub-daily climate variables from daily data based on profiling the diurnal pattern of the different variables using historical data. Days of meteorology similarity were selected based on different daily summaries of selected variables on which the diurnal patterns were generated. The model was developed and finetuned by adjusting the window size of DOY. Overall, I think the methodology presented by this paper has a big drawback which the authors have not fully understood to be able to explain and present well. The structure of and the reasoning in this manuscript is not well designed as a qualified academic journal paper. The English of this manuscript need a significant improvement. I would require a native English speaker to review the entire manuscript before submitting. My comments are listed below.

**Reply: Thanks a lot for your feedback. We improved the language throughout the whole manuscript and completely revised the structure of the paper, dividing it into an introduction, data, methods and results section.**

- The manuscript is not structured as an academic journal paper. There is no section of "method", "results", and "conclusion". The introduction did not explain well the necessity to develop the teddy tool against the current practices and methods. Why is Teddy better than others?

**Reply: We completely revised the structure of the paper by including a data, method and results section. Thanks a lot for this note, we agree that this improved the quality and readability of our paper. Please read line 94-103 (of the revised manuscript), in which we describe the uniqueness of the Teddy tool in comparison to other approaches. Please be aware that we do not claim that our approach is better than other approaches. The unique features of our method are conservation of mass and energy, physical consistency of inter-variable relationship, application of empirical diurnal profiles, choice of the daily climate analogue based on an equal ranking of all available climate variables. The choice of the disaggregation method and its evaluation as "better" or "more appropriate" depends strongly on the use case.**

- There is no "data" section in this manuscript that specifically talk about the source and quality of data. Such information is distributed all over the manuscript which makes it hard to read.

**Reply: We fully agree, that a data section would help to better understand what data sources are required. We inserted a section 'Data and data requirements' right after the introduction (see line 124-248) and shifted Table 1, which gives an overview of used climate variables and datasets into this section. We added a description of the WFDE5 data, which is here used as an hourly example reference dataset. However, the Teddy-Tool can generally be applied on any daily climate data with an associated hourly reference data set.**

- The "discussion" section reads more like a summary of the manuscript instead of giving the reasoning of the results and the corresponding performance impacts on this method. For example, the results of precipitation and wind speed is not as good as other variables. Why did this happen? Is it the limits of the method? What are the potential solutions based on the reason?

**Reply: Thanks a lot, we revised the discussion section and extended the description of methodological limitations. As suggested, we also added an explanation why results of precipitation and wind speed is not as good as other variables (see line 560-563). We refer this to the high intra-day variability of rainfall and wind speed (Watters et al., 2021). Other variables are governed by a stronger daily cycle (Dai and Trenberth, 2004), which is easier to disaggregate based on diurnal profiles. In line 580-649, we discuss limitations of the method. However, we would not state that the lower performance for rainfall and wind speed disaggregation is caused by the limits of our method but by internal climate variability inducing high intra-day variability. There is no perfect solution as internal climate variability is intrinsic to the system. We tried to improve the performance of Teddy for precipitation and wind speed disaggregation by introducing the DOY window (see Fig. 5).**

- The "similar meteorology day" is defined in a statistical way. What is the meteorology evidence on which such definition is built? Why pick those parameters for consideration? The descriptions of the ranking and the hourly profile is very vague for me to understand.

**Reply: As you correctly say, we define the 'most similar meteorological day' in a statistical way. Therefore, we pick all available daily variables from the bias-corrected climate models (tas, tasmax, tasmin, pr, hurs, rsds, rlds, ps, sfcwind) and compare it with the variables of the hourly WFDE5 dataset, which has been aggregated to daily values for this comparison. The available data is explained and described in the newly added data section. Since the absolute or relative errors for the different meteorological variables cannot be simply compared to each other to assess the similarity between meteorological days, we implemented a ranking approach, which allows to equally weight all considered variables for this comparison. In this context, we define 'the most similar meteorological day' as the day with the minimum sum of ranks. Thus, the 'most similar meteorological day' refers to the statistical similarity of selected near-surface meteorological variables at a given location and time. The approach works under the assumption that similar daily values would have a similar sub-daily profile (which is also assumed by the method of fragments, see: Li et al., 2018; Pui et al., 2012; Sharma et al., 2006). In order to avoid misunderstandings, we revised the description in the methodology section and included a clear definition of the 'most similar meteorological day'. In addition, we included an explanation, why we chose the ranking approach (see line 324-331).**

- I totally got lost by the sentence at Line 140. To aggregate to what time step? What do the numbers represent?

**Reply: Thank you for this comment. We agree that this was difficult to understand. The sentence should address that the user is able to adjust the temporal resolution of the Teddy output, which can be set to 1-, 2-, 3-, 4-, 6-, 8-, or 12-hourly values. We revised this sentence (see line 346) and included a sentence in the data section, where the temporal resolution of the different input data and output data is described (see line 147-148).**

- The processing of precipitation in rare cases was not explained well. Meteorologically, nighttime distribution is not valid. I was curious why solar radiation is very important in determine the night time distribution. No citation or justification provided.

**Reply: In case that no historical information about the hourly distribution of precipitation is available, we still have to distribute precipitation according to the premise that mass and energy are conserved throughout the disaggregation procedure. Please keep in mind that this happens only in very few cases in extremely dry deserts, where WFDE5 data show no precipitation event within the last 40 years. We added Supplementary Figure S1 to show pixels where this is the case. A goal of Teddy is to consider the physical consistency of inter-variable relationships. Precipitation generally affects other climate variables (e.g. humidity, radiation, temperature, etc.; Meredith et al., 2021). During night, physical interdependencies between precipitation and other variables are generally lower, because radiation is not affected and less energy is available to affect other variables. This might have an effect for impact models, because, as an example, evapotranspiration might be unrealistically high if precipitation occurs at the same time with full solar irradiation during noon. Therefore, in case no information is available, we restrict precipitation events to hours at nighttime. Since 'nighttime' might be defined differently, we refer to the time between sunset and sunrise. Accordingly, this varies between different locations and seasons. However, we agree that the assumption might not be valid for all user cases as we agree that it is not meteorologically valid. Therefore, we now additionally implemented the option to write Not a Number (NaN) values instead. We revised and extended the description of the processing of precipitation according to your comment (line 394-416).**

- The methodology of the paper is based on the profiling of the daily fluctuation of different variables which is built on the autocorrelation of adjacent time step. However, this assumption holds nicely on continuous variables but not the discrete ones, such as precipitation and wind speed. There is no discussion of this restriction and no solution to address that.

**Reply: We are not sure if we understand your comment correctly. We interpret your comment as addressing the limitations of inter-day connectivity (or "adjacent time step"). However, to our understanding, precipitation and wind speed are continuous variables, bounded at zero. This limitation applies to all variables at varying degree, however to a greater extent for precipitation and wind speed (as you also argue). Therefore, we additionally included Supplementary Figure S2 (similar to Fig. 3) for sample location 22 in China to visualize the limitation of the approach regarding the inter-day connectivity. In Fig. S2, it is visible, that e.g. between July 1$^{st}$ and July 2$^{nd}$, several variables (e.g. tas and hurs) show a prompt decline. We touched upon this issue in the discussion in the last paragraph, and we are thankful for your critique to discuss this issue in more detail. Generally, possible jumps between the days for the disaggregated time series are also seen in other disaggregation approaches, such as the method of fragments. Hence, for precipitation, we additionally included the consideration of the precipitation status (dry or wet) for the day before, after, and the day of interest. In the discussion section, we added a paragraph on this limitation (lines 588-598) and extended the discussion of possible improvements (lines 650-658).**

- Many of the efforts in this paper have been put into adjusting the size of window. I was wondering why the author think this is such an important variable that may

influence the results? If this is used for finetuning, then it is more critical to present the finetuning procedure instead of the results.

**Reply: We hypothesized that the size of the moving window has a significant impact on the results, since it determines the sample size. We could show that the size of the moving window is a relevant parameter for such an approach, affecting especially the evaluation results for precipitation and wind speed. This also refers back to your remark, what "solution" can be provided for the lower performance of the disaggregation of precipitation and wind speed. The tuning of the window size showed at least minor performance improvements (see Figs. 5-8). Furthermore, the window size is a methodological parameter, which can be set by the user. Apart from that, our method is non-parametric. Hence, we wanted to show its influence on the resulting disaggregation as an overview for potential users.**

Dai, A., and K. E. Trenberth, 2004: The Diurnal Cycle and Its Depiction in the Community Climate System Model. *J. Climate*, **17**, 930–951, https://doi.org/10.1175/1520-0442(2004)017<0930:TDCAID>2.0.CO;2.

Watters, D., A. Battaglia, and R. P. Allan, 2021: The Diurnal Cycle of Precipitation according to Multiple Decades of Global Satellite Observations, Three CMIP6 Models, and the ECMWF Reanalysis. *J. Climate*, **34**, 5063–5080, https://doi.org/10.1175/JCLI-D-20-0966.1.

Pui, A., Sharma, A., Mehrotra, R., Sivakumar, B., and Jeremiah, E.: A comparison of alternatives for daily to sub-daily rainfall disaggregation, J. Hydrol., 470, 138– 157, https://doi.org/10.1016/j.jhydrol.2012.08.041, 2012.

Sharma, A. and Srikanthan, S.: Continuous Rainfall Simulation: A Nonparametric Alternative, in: 30th Hydrology & Water Resources Symposium: Past, Present & Future, 4– 7 December 2006, Launceston, Tasmania, p. 86, 2006.

Meredith, E.P. *et al.* 2021 *Environ. Res. Commun.* **3** 055002 https://doi.org/10.1088/2515-7620/abf15e

Dear Reviewer #3,

thanks a lot for your valuable and helpful comments that are much appreciated and helped us to improve our paper. In the following, we reply to your general and specific comments. Thereby, our reply is placed right below your comment and marked bold.

The manuscript **on** "The Teddy-1 Tool v1.0: temporal disaggregation of daily climate model data for climate impact analysis" **is** indeed the utmost work that can be published for closing the gap of data limitation. The authors have done amazing work. However, I have found the manuscript is poorly written, described, and analyzed. I like to see an additional section for uncertainty analysis. This will improve the manuscript. Currently, the manuscript is unfit for publication and I would recommend it with a major change.

**Reply:** Thanks a lot for your feedback. It's great to see that there is need for such a tool. Based on your comments and the comments of the other reviewers, we were able to improve the manuscript and also extended the analysis a lot. We added subsections to the Result sections and also included several uncertainty assessments.

 Teddy-Tool is poorly performed in precipitation generation which is one of the most important parameters due to its intermittency. The authors failed to use other statistical parameters to evaluate the performance of the tool. Using a coefficient of correlation merely confirms the satisfactory performance while disaggregating the various parameters. In addition to providing model biasness, I encourage to provide the model error (e.g., MAE and RRSE). NSE and Coeff. of determination are encouraged to include as well for all of the parameters.

**Reply:** We agree and therefore included MAE, RMSE, and NSE to Figure 3 and Figure 4 (as well as Figure S2 and Figure S3) to better describe the model performance. In addition to Fig. 5, we added Supplementary Fig. S4 showing the NSE for each doy-window size and also Supplementary Figure S5, showing the MAE for each year.

Moreover, the authors are encouraging to analyze the qualities of datasets. For instance, precipitation characteristics include the number of dry/wet days, annual amounts, and extreme values.

**Reply:** The number of wet/dry days and annual amounts are not relevant for the disaggregation performance, as Teddy by definition reproduces daily precipitation sums (and therefore also annual sums). Nevertheless, we added Figure 10, showing the number of wet ours per month.

Fig. 9 used GEV for comparing the annual maxima with bootstrapping resampling technique. The manuscript didn't properly mention how parameters were estimated.

**Reply:** Thank you for this hint, we added a sentence on the estimation and goodness-of-fit test.

Also, I like to see whether is there any significant difference between the reference and disaggregated hourly datasets. Furthermore, I like to see a plot of rainfall intensity versus percent exceedance curves between reference and reference and disaggregated hourly datasets (See Fig 10. Choi et al 2008- Hourly Disaggregation of Daily Rainfall in Texas Using Measured Hourly Precipitation at Other Locations).

**Reply:** We added a very similar figure, where Teddy shows overall good performance in the reproduction (except for extremes, which was diagnosed already).

**Comments:**

1.Introduction

I am hoping to include or discuss a relevant recent literature review as indicated in L62-63.

**Reply:** We added Pui et al. (2012) here.

L44-46- reference?

**Reply:** We added Juckes et al. (2020) and Luttgau & Kunkel (2018).

Why do you choose the statistical method over the others?

**Reply:** Other approaches, such as a dynamical downscaling is much more elaborate and expensive in regards to computational demand, complexity and working time.

L62-63- why?

**Reply:** Empirical approaches might e.g. use specific relationships, such as e.g. the relationship between precipitation intensity and precipitation duration that are taken for a specific region and not from a globally available dataset for a specific location. Also, relationships between specific variables or that are not applicable in other climate zones or between northern and southern hemisphere might result in inconsistencies if the model is not designed to be globally applicable.

L63-66- reference?

**Reply:** added

L67-68 – I believe mass and energy conservation is part of the procedure for disaggregation of the climate parameters. I cannot grasp the relationship with the previous sentence.

**Reply:** Not all disaggregation strategies (weather generators and dynamical downscaling) conserve mass and energy of the global climate model. Here, we wanted to clarify that we conserve mass and energy, which is relevant for the further use of impact models.

L71-76 Please split into multiple sentences. This is confusing with long sentences. It will help the readers like me with comprehension.

**Reply:** Thank you, done.

**2. Temporal Disaggregation:**

The method needed to include an uncertainty analysis associated with the method (as mentioned above). Using the mean of variables except precipitation may have led to the underprediction of extreme values as in Fig 3 (I recommend avoiding colors in Fig.3 as long as we can use different line types).

**Reply:** We tried to use line types instead of colors, however due to the large overlap, the readability of the figure is much better by using colors. We added transparency to the lines, which helped improving the legibility. We added a section on precipitation extremes.

Is the climate model data bias corrected? If so, why do you have a drizzle effect or bias as mentioned in L116?

**Reply:** Climate model data is bias adjusted with reanalysis data and hence still contains drizzle (Lange 2019).

What is AE in Fig 1? Is it an absolute error (AE) at L123?

**Reply:** Yes, thank you, this is the absolute error, we added the abbreviation in line 123!

What is the threshold for selecting the min value?

**Reply:** There is no threshold.

Does the method compensate for any parameters having unsatisfactory AE values?

**Reply:** We thought about that when designing the Teddy-Tool. However, thresholds for "unsatisfactory" AE values are arbitrary and we decided to not implement them.

Moreover, the manuscript didn't discuss it. I like to see a comprehensive result and discussion about the method and findings. This is one of the important sections of the manuscript.

**Reply:** We extended the evaluation and the discussion section, also according to the other reviewer comments.

L84- check the numbering of the heading.

**Reply:** According to other reviewer comments, **w**e added several new subheadings and sections. Therefore, we revised the heading completely.

L91/ L72- what are those periods?

**Reply:** ISIMIP provides data for historical time periods (1850-2014) and future time periods (2015-2100) for different scenarios (SSP126, SSP370, SSP585).

L97- How does this procedure help in minimizing computational resources? I don't understand the specific statement. Isn't it the procedure part of the disaggregation?

**Reply:** The precalculation computes daily averages from the hourly reference dataset. This saves computational resources and computational time, because this is only calculated once and stored in files for further use.

L112- what is the significance/logic of choosing 11?

**Reply:** We performed a sensitivity analysis for choosing different DOY window sizes. In section 'Sensitivity analysis DOY window size', we discuss your question.

L114- precipitation state?

**Reply:** Thank you, (wet/dry) added.

L129-130 I am lost with the meaning of daily mean value from the climate model. L142 didn't help me either. Are you referring to that precipitation used aggregated value not mean? This seems like multiplication. Please avoid this confusion.

**Reply:** By adding the hourly profile to the mean value, by maintaining the mean, we guaranty energy and mass conservation. For precipitation, we use the sum, not the mean. We changed the manuscript accordingly.

L140-141 units?

**Reply:** added

L145- I would prefer to use "no" instead of failing.

**Reply:** done

L147-149- again how do you choose the window period? Is it depending on the user or the available referenced datasets? What is the rationale for using linear regression between the duration and precipitation amount? Is this entirely a novel approach? If not provide references.

**Reply:** We added Fig. S1 in the Supplement to show, where this issue occurs. The approach to handle the issue of no precipitation in the reference data is a compromise. The increase of the window size to +-50 days is arbitrary. As this approach is a compromise, we also added the option for the user to write Not a Number (NaN) values instead. We don't know any other approach that takes this into account.

---

## Editor Decision (ED1)

**Referred to Track changes' Manuscript:**

Teddy-Tool v1.1 performed satisfactorily for all variables except precipitation and windspeed. Considering the limitations of the methodology, I recommend a minor revision. I encourage authors to include the following comments and suggestions.

**Comments & Suggestions:**

a) Comparison of absolute error (AE) of each variable between the reference/observed and climate model datasets seems not enough for choosing "the most similar meteorological day". For instance, AE compared only the rainfall amount between daily reference/observed and climate model datasets, but it ignores the characteristics such as intensity and frequency such as heavy (in ISIMP) and light (aggregated daily WFDE5) rainfall events.

b) How do authors justify with a minimum sum of ranks has the "most similar meteorological day"?

L134 add the sentence to the previous paragraph. It's confusing with Table 1.
L223-228- justify. Why filtration of statistical population based on precipitation state based on three days (i.e., 8 options)?
L232- Why do we consider an equal weightage for each variable? Does the correlation among each variable as indicated in L345-346 have any significant effects on it?

L234- elaborate on the term statistical similarity meant.
L508-509- paraphrase
Fig1: check whether the chart needs to include the portion of L305-307
Fig: 9- r value between which parameters? (Same as Fig 5 & 6)
Table 1- check the subscripts & include the full name of variables such as Precipitation (pr)

- include the ISIMIP data description in the manuscript's data section as provided in the response section.
- Rename the 4.3 section
- L562- "Discussion and summary" seems appropriate.

---

## Author Response (AR2)

**Referred to Track changes' Manuscript:**

Teddy-Tool v1.1 performed satisfactorily for all variables except precipitation and windspeed. Considering the limitations of the methodology, I recommend a minor revision. I encourage authors to include the following comments and suggestions.

**Reply: Thanks a lot for your detailed further comments that we appreciated very much!**

**Comments & Suggestions:**

1. a) Comparison of absolute error (AE) of each variable between the reference/observed and climate model datasets seems not enough for choosing "the most similar meteorological day". For instance, AE compared only the rainfall amount between daily reference/observed and climate model datasets, but it ignores the characteristics such as intensity and frequency such as heavy (in ISIMP) and light (aggregated daily WFDE5) rainfall events.

**Reply: Using the absolute error to rank the daily variables is a well justified approach, which is also applied within the 'Method of Fragments' (Li et al., 2018; Pui et al., 2012; Sharma et al., 2006). We emphasize that the choice of error metric (e.g. AE or MSE/RMSE) does not play a role, as the ranks of errors are the same for AE and (R)MSE.**
**Intensity and frequency characteristics of rainfall are consistent between WFDE5 and ISIMIP, as these characteristics have been aligned within the bias adjustment (Lange 2019). Hence, we do not have to compare overall characteristics of rainfall or other variables. We clarified this within the text: "The bias-adjusted hourly WFDE5 data is globally available for the time period between 1979 and 2019 at 0.5° spatial resolution. It is consistent with the bias-adjustment procedure within ISIMIP (Lange, 2019) and thus provides a consistent hourly reference data for Teddy."**

2. b) How do authors justify with a minimum sum of ranks has the "most similar meteorological day"?

**Reply: This is a defined assumption of this approach (see line 330: "In this context, we define 'the most similar meteorological day' as the day with the minimum sum of ranks"). We add references for justification: "The approach works under the assumption that similar daily values would have a similar sub-daily profile (Li et al., 2018; Pui et al., 2012; Sharma et al., 2006)". As shown by the evaluation results, this assumption can be regarded as well suitable for many applications.**

L134 add the sentence to the previous paragraph. It's confusing with Table 1.

**Reply: Thank you. Done.**

L223-228- justify. Why filtration of statistical population based on precipitation state based on three days (i.e., 8 options)?

**Reply: The main reason for considering the precipitation state of the contiguous days is to improve the inter-day connectivity. Without this filtration there would be an increased probability to "cut off" rainfall events between days. This would induce jumps for rainfall (but also other connected variables) between days, as these can be chosen**

**from different time periods. Therefore, we implemented this filtration step as also suggested by Li et al. (2018) and Poschlod et al. (2018). This justification is reflected in the text: "This step is included to better reproduce the inter-day connectivity of precipitation (Li et al., 2018)".**

L232- Why do we consider an equal weightage for each variable? Does the correlation among each variable as indicated in L345-346 have any significant effects on it?

**Reply: We weight all variables equally because we are interested in an overall representation of the meteorological conditions for the corresponding day. Generally, a user could modify the weightage dependent on the specified further use. However, in this general model description, we would not want to imply different importance of variables.**

L234- elaborate on the term statistical similarity meant.

**Reply: We use a statistical approach. This was meant by 'statistical similarity'. We change the term to "statistically derived similarity" in order to emphasize that our approach considers the similarity in a statistical manner. The methodological elaboration on this statistically derived similarity is given in the previous lines (L201 – 234). The justification of this assumption is given by the references to Li et al., 2018; Pui et al., 2012; Sharma et al., 2006 in L235/302 directly thereafter.**

L508-509- paraphrase

**Reply: Done.**

Fig1: check whether the chart needs to include the portion of L305-307

**Reply: Thanks a lot, we included that to the chart in Figure 1 (hourly value = hourly profile * daily mean value of the climate model).**

Fig: 9- r value between which parameters? (Same as Fig 5 & 6)

**Reply: Thank you for indicating this, we added this information to the figure caption of Fig. 5, Fig. 6, Fig. 7 and Fig. 9.**

Table 1- check the subscripts & include the full name of variables such as Precipitation (pr)

**Reply: We added the full names of the variables to Table 1.**

- include the ISIMIP data description in the manuscript's data section as provided in the response section.

**Reply: We included the ISIMIP data description in the manuscript's data section as provided in the responses.**

- Rename the 4.3 section

**Reply: Done.**

- L562- "Discussion and summary" seems appropriate.

**Reply: Thank you!**